# Characterization of Contractile Proteins from Skeletal Muscle Using Gel-Based Top-Down Proteomics

**DOI:** 10.3390/proteomes7020025

**Published:** 2019-06-20

**Authors:** Paul Dowling, Margit Zweyer, Dieter Swandulla, Kay Ohlendieck

**Affiliations:** 1Department of Biology, Maynooth University, Maynooth, W23F2H6 Co. Kildare, Ireland; paul.dowling@mu.ie; 2MU Human Health Research Institute, Maynooth University, Maynooth, W23F2H6 Co. Kildare, Ireland; 3Institute of Physiology II, University of Bonn, D-53115 Bonn, Germany; margit.zweyer@ukbonn.de (M.Z.); dieter.swandulla@ukbonn.de (D.S.)

**Keywords:** actin, mass spectrometry, muscle proteomics, myosin, top-down proteomics, tropomyosin, troponin, two-dimensional gel electrophoresis, skeletal muscle

## Abstract

The mass spectrometric analysis of skeletal muscle proteins has used both peptide-centric and protein-focused approaches. The term ‘top-down proteomics’ is often used in relation to studying purified proteoforms and their post-translational modifications. Two-dimensional gel electrophoresis, in combination with peptide generation for the identification and characterization of intact proteoforms being present in two-dimensional spots, plays a critical role in specific applications of top-down proteomics. A decisive bioanalytical advantage of gel-based and top-down approaches is the initial bioanalytical focus on intact proteins, which usually enables the swift identification and detailed characterisation of specific proteoforms. In this review, we describe the usage of two-dimensional gel electrophoretic top-down proteomics and related approaches for the systematic analysis of key components of the contractile apparatus, with a special focus on myosin heavy and light chains and their associated regulatory proteins. The detailed biochemical analysis of proteins belonging to the thick and thin skeletal muscle filaments has decisively improved our biochemical understanding of structure-function relationships within the contractile apparatus. Gel-based and top-down proteomics has clearly established a variety of slow and fast isoforms of myosin, troponin and tropomyosin as excellent markers of fibre type specification and dynamic muscle transition processes.

## 1. Introduction

Mass spectrometry (MS) based proteomics is concerned with the systematic analysis of the protein components of a particular biological fluid, cell type, tissue, organ or organism [1], and plays a key role in protein biochemistry and molecular cell biology [2]. Modern proteomics is based on high-performance mass analysers [3,4,5] and is routinely used for cell-specific protein expression profiling, large-scale comparative protein studies, the biochemical characterization of the composition and structure of individual subcellular proteomes, and the determination of the temporal and spatial dynamics within specific protein constituents [6,7,8,9]. Gel-based proteomics is one of the most multipurpose means for the analysis of complex proteomes, such as seen in skeletal muscle tissues [10,11,12]. Two-dimensional gel electrophoresis (2DGE) exemplifies an exceptionally robust strategy to separate, identify, and quantify proteins when used in combination with MS analysis or various immunological tests [13,14,15]. While 2DGE is an effective technique for the systematic analysis of complex protein lysates, the ability to directly visualise changes in the abundance of proteins/post-translational modifications (PTM) is a powerful application.

Since 2DGE approaches present a visual platform for the efficient separation of individual proteoforms prior to MS analysis, the availability of internet-based comparisons via international 2D-gel databases is a strength of gel-based proteomics. A considerable number of 2DGE databases were established, especially in the period 1993–2008 [16,17,18,19,20,21,22,23,24,25,26,27,28], and include SWISS-2DPAGE, World-2DPAGE, Human 2-D PAGE biobase, Flicker, UAB Proteomics Database, AGML Central, GELBANK, WEB P.A.G.E, Open2Dprot Project, LECB 2-D PAGE Gel Images Data Sets, ProteomeWeb, PHProteomicsDB, pProRep and DynaProt 2D. These extensive collections of 2DGE images and bioinformatic analysis tools can be highly useful for the comparative analysis of newly established 2DGE images, the systematic cataloguing of proteoforms using 2DGE techniques and the integration of gel-based proteomic findings and mass spectral data [28]. Comparative image analyses can also be helpful for the swift identification of one-dimensional (1D) protein bands and 2D protein spots within archived gels that were produced prior to the establishment of MS-based proteomics [29,30]. Unfortunately, many of these 2DGE image databases are no longer actively maintained, which constitutes a drawback in the further development of the field of gel-based proteomics and hampers the efficient sharing of proteomic data generated by 2DGE analyses. Hopefully, international efforts will be able to re-establish functional and internet-based databanks for future gel electrophoretic studies and the efficient comparison of proteomic maps.

Enormous progress has been made in the establishment of the tissue-based map of the human proteome [31], including the systematic cataloguing of the protein constituents of skeletal muscles [32,33,34,35]. Large-scale biochemical studies with a focus on the protein composition of voluntary contractile tissues were carried out by various proteomic techniques with a peptide-centric approach following the digestion of muscle proteins extracted from various animal species prior to peptide MS analysis [10,11,12]. These proteomic surveys have investigated in detail the complex effects of fibre transitions, neuromuscular disease, exercise, disuse atrophy and aging on the skeletal muscle proteome [36,37,38,39]. Efficient muscle protein separation by liquid chromatography (LC) or diverse gel electrophoretic techniques have been employed in combination with in-solution or in-gel digestion protocols, whereby 2DGE plays a key role in proteome science [13,14,15,40,41]. The bioanalytical advantages versus potential technical limitations of gel electrophoretic approaches, as applied to the highly complex and dynamic proteome of skeletal muscle tissues, have been previously discussed in an extensive review on comparative skeletal muscle proteomics using 2DGE [42]. The recent introduction of the photo-cleavable anionic surfactant 4-hexylphenylazosulfonate as a potential sodium dodecyl sulfate replacement in polyacrylamide GE [43] might be suitable for improved gel-based and top-down proteomic workflows [44].

Using a variety of GE- and LC-based protein separation techniques in combination with MS analysis, over 10,000 muscle-associated proteins have been identified [32,33,34,35,45,46,47,48]. Since peptide-based and bottom-up bioanalytical methods can often not properly distinguish among discrete proteoforms, such as individual isoforms or post-translationally modified protein variants, biochemical approaches using top-down proteomics with a focus on the mass spectrometric characterization of intact protein species can be advantageous [49,50,51]. Importantly, specific forms of top-down proteomics can provide data for accurate protein mass determination and detailed information on the identity of specific protein isoforms and their PTMs [52,53,54], making it an ideal investigative tool of modern protein biochemistry for determining the structure and function of macromolecular protein complexes [55,56,57]. In this review, the usage of top-down proteomics in the field of skeletal muscle biochemistry is outlined with special emphasis on gel-based techniques, especially 2DGE, for studying normal, adapting and aging muscles. Detailed descriptions and critical assessments of the establishment of the skeletal muscle proteome and the usage of comparative MS-based proteomics for studying pathobiochemical changes in neuromuscular disorders have already been extensively reviewed [10,11,12,58,59,60].

The contractile function of the skeletal muscle system is based on the dynamic arrangement of an unusual cell type that is very large, multi-nucleated and extremely heterogeneous in its protein isoform composition [61]. A striking biochemical feature of contractile tissues is the presence of a considerable number of very high-molecular-mass protein species [62], which are occasionally difficult to separate for proteomic studies [63]. The physiological coupling of individual muscle fibres to its innervating motor neuron within distinct motor units influences skeletal muscle mass, fibre size, contractile kinetics, energy metabolism, muscle cell specificity and fibre type-related protein expression pattern [64]. Besides a mixture of fast-twitching, slow-twitching and hybrid fibres [65], individual skeletal muscles also contain several layers of connective tissue, capillaries and myospecific stem cells. Although this heterogeneous cellular composition often complicates the systematic mass spectrometric profiling of the skeletal muscle proteome in health and disease [66], skeletal muscle proteomics has clearly confirmed the multi-functionality and high adaptive capacity of the voluntary contractile system. This brief review on representative top-down proteomic studies of contractile proteins from skeletal muscle fibres illustrates how the systematic application of gel-based and MS approaches has improved our biochemical knowledge of protein structure-function relationships and the dynamic nature of the neuromuscular proteome.

## 2. Top-Down Proteomics

Top-down proteomics is an advanced biochemical method for the quantitative analysis of specific proteoforms leading to the identification and thorough characterization of intact protein species by tandem MS/MS analysis. Fundamental differences in the analytical approach used in top-down proteomics, as compared to middle-down proteomics or bottom-up proteomics are described in Figure 1.

Comprehensive guides for sample preparation and discussions of technical issues in relation to specific approaches in top-down protein analysis have been published [67,68,69,70,71,72]. The main analytical advantages of this approach are the determination of distinct characteristics of a particular protein isoform and the capability to distinguish degradation products, as well as identify sequence variants and combinations of dynamic PTMs [73].

While top-down proteomics focuses on the MS analysis of intact proteins with a proteoform-centric approach, the other proteomic strategies involve the study of differently sized protein fragments following protein digestion and peptide generation. Middle-down proteomics is used for the analysis of large protein fragments and the most frequently employed bottom-up proteomics approach is a peptide-centric technique [74]. In this article, we define ‘top-down proteomics’ in its broadest sense and use the term not just in a focused way describing exclusively the MS/MS analysis of an intact protein, but also include the characterization of distinct proteoforms using gel-based separation technology in combination with routine proteomic identification methods. This review includes therefore bioanalytical approaches that initiate the proteomic analysis with a separated and thus isolated proteoform of a particular protein and then employ differing MS methods for protein/peptide identification and/or PTM analysis.

Bioanalytical advantages versus potential technical limitations of gel-based top-down proteomics, especially those issues associated with 2DGE approaches [42], are listed in Figure 2.

Gel-based methods are robust, highly reproducible and reliable protein separation techniques that are ideally suited for large-scale surveys of proteoforms. Staining of gel electrophoretically separated proteins can be achieved with a wide range of highly sensitive dyes, such as colloidal Coomassie Brilliant Blue, silver stains and various fluorescent dyes, as extensively reviewed [75,76,77,78,79,80]. The analytical strength of gel-based methods, as compared to protein separation by LC techniques, is the direct visualization of proteins within 1D gel bands or 2D gel spots. Thus, the position of discrete protein spots in 2D gels provides a swift and reliable way for determining the characteristic combination of the relative molecular mass and the p*I* (isoelectric point) value of a specific proteoform of interest. Potential bioanalytical complications of gel-based top-down proteomics are associated with the fact that prolonged periods of sample preparation and complex isolation steps may introduce protein modifications prior to MS analysis. Routine 2DGE analyses may also under-estimate the presence of certain types of proteins, such as high-molecular-mass proteins, very hydrophobic proteoforms and low-abundance proteins [42].

## 3. Top-Down Proteomics of Contractile Proteins from Skeletal Muscle

### 3.1. Acto-Myosin Apparatus and Auxilary Components

In the longitudinal direction, mature skeletal muscle fibres exhibit a characteristic striated appearance due to the regular patterning of anisotropic A-bands and isotropic I-bands, which form sarcomeric units between two neighbouring Z-disks [81]. A major feature of muscle genetics is the fact that the isoform-specific expression of a limited number of genes that encode contractile components leads to the production of the majority of the protein content of contractile cells [61]. Within the sarcomere structure, the contractile apparatus of skeletal muscle fibres consists of two main contractile filaments forming the acto-myosin system [82,83,84] and two auxiliary filamentous systems acting as myofibrillar gap structures [85,86,87,88]. The complex network of the sarcomere units consists mainly of myosin light and heavy chains, actin and the regulatory tropomyosin/troponin complex, the giant proteins titin and nebulin, as well as the structural and functional support units presented by the M-band zone and the Z-disk complex [89].

The principal motor molecules of the contractile protein assembly, which produce force via cross-bridge/swinging lever-arm mechanisms [90], are filamentous actin and the myosin heavy chains [91]. The entire myosin complex exists as a hexameric structure of two myosin heavy chains (MyHC) and four myosin light chains (MLC) [92]. The MyHC head structure mediates the reversible coupling and cross-bridging process between the thick filaments located in the A-band region and the actin filaments [93]. The regulatory and catalytic light chains provide critical phosphorylation sites, which are essential for the movement of phosphorylated myosin cross-bridges away from the thick filament, as well as the fine tuning of myosin motor function by providing structural stability to the lever arm domain of the myosin head [94]. Below listed are the main components of the thick myosin-containing filament, the thin actin-containing filament, the auxiliary titin filament, the auxiliary nebulin filament, the sarcomeric M-band complex and the sarcomeric Z-disk complex [66,95]:**Thick myosin-containing filament**: hexameric myosin complex consisting of myosin heavy chains (MyHC I, IIa, IId/x, and IIb), myosin light chains (fast and slow MLC isoforms; regulatory and catalytic subunits) and myosin binding proteins**Thin actin-containing filament**: α-actin, α/β-tropomyosin, troponin complex (fast and slow troponin-I, troponin-T and troponin-C isoforms)**Auxiliary titin filament**: half-sarcomere spanning titin and muscle ankyrin repeat protein**Auxiliary nebulin filament**: nebulin, in close contact to actin-containing thin filament**Sarcomeric M-band complex**: myomesin and obscurin, linked to titin filament**Sarcomeric Z-disk complex**: α-actinin, plectin, telethonin, desmin, myozenin, myotilin, synemin and filamin

The systematic 2DGE separation of protein isoforms from skeletal muscle subtypes has led, in combination with MS analysis, to the identification of distinct isoforms of myosins, actins, troponins and tropomyosins, as well as many associated proteins of the sarcomeric structure [10,42]. Building on the findings from initial gel-based studies of the skeletal muscle proteome [96,97,98], high-resolution 2DGE was applied to the cataloguing of human *vastus lateralis*, *deltoideus* and laryngeal muscles [99,100,101,102] and a variety of animal species including rat, rabbit, cow and fish [103,104,105,106,107,108,109,110]. The differential expression patterns of contractile proteins in predominantly fast- versus slow-twitching skeletal muscles was established for various muscle subtypes, such as *gastrocnemius*, *extensor digitorium longus*, *longissimus dorsi*, *semitendinosus* and *soleus* muscles, using 2DGE [111,112,113,114,115]. The gel-based fibre specification maps agree with the distribution of fast versus slow muscle protein isoforms as determined by *in vivo* stable isotope labelling with amino acids in the SILAC mouse model [116]. Figure 3 shows representative 1DGE and 2DGE images of separated muscle protein populations. These approaches are routinely used for the proteomic identification of distinct proteoforms of contractile components.

### 3.2. Top-Down Proteomics of Myosins

Muscle fibre type classification can be conveniently carried out by determining the distribution of myosin light and heavy chain isoforms, making this large family of contractile proteins an essential class of fibre type markers [92]. The expression pattern of MyHC molecules is often performed by immunofluorescence microscopy of transverse cryosections from individual skeletal muscles [117]. Most adult skeletal muscles contain a mixture of the main fibre types I, IIa, IIx, IIb, as well as the mixed hybrid fibre types I/IIa, IIa/IIx and IIx/IIb. However, considerable species-specific differences exist in relation to the distribution of cellular subtypes within large fibre populations [118]. Although at least 11 MyHC isoforms and nine MLC isoforms appear to exist in developing and matured skeletal muscles [119,120], the spectrum of slow-twitching oxidative fibres towards faster-twitching and more glycolytic fibres can be accurately evaluated by the abundance of the main human slow/cardiac MyHC-I-beta (P12883; *MYH7* gene), fast MyHC-2a (Q9UKX2; *MYH2* gene), fast MyHC-2x (P12882; *MYH1* gene) and fast MyHC-2b (Q9Y623; *MYH4* gene) isoforms [117,118]. In addition, developing and regenerating muscles contain MyHC-neo (P13535; *MYH8* gene) and MyHC-emb (P11055; *MYH3* gene), and the highly specialist extraocular muscle fibres are characterized by the presence of MyHC-EO (Q9UKX3; *MYH13* gene), MyHC-I-ton (A7E2Y1; *MYH7b* gene) and MyHC-15 (Q9Y2K3; *MYH15* gene) [92]. Human MLC molecules exist mainly as essential catalytic subunits and regulatory/phosphorylatable subunits, i.e., as fast MLC1/MLC3 (P05976; *MYL1* gene) and fast MLC2 (Q96A32; *MYLPF* gene) or slow MLC1 (MLC3) (P08590; *MYL3* gene) and slow MLC2 (P10916; *MYL2* gene) isoforms. The combination of various MyHC and MLC subunits form a very large number of hexameric iso-myosins in the sarcomere [92,121,122]. In contrast, human filamentous F-actin polymers in developing and mature fibres consist of only two G-actin isoforms, i.e., cardiac alpha-actin (P68032; *ACTC1* gene) and muscle alpha-actin (P68133; *ACTA1* gene). An overview of the systematic characterization of contractile proteins from skeletal muscle using top-down proteomics is provided in Figure 4.

The detailed proteomic analysis of fast versus slow skeletal muscles has confirmed the complexity of myosin expression patterns. Due to their differing molecular mass and electrophoretic mobility, heavy chains and light chains of myosin complexes are routinely studied by 1DGE versus 2DGE, respectively. The 2DGE top-down proteomic analysis of rat *gastrocnemius* versus *soleus* muscle extracts has clearly shown the differential expression pattern of myosin light chains MLC1s and MLC2s in slow muscle versus MLC1f, MLC2f and MLC3f in fast muscle [114]. Thus, myosin subunits represent reliable proteomic markers of fibre type specification and the expression of the main fast versus slow isoforms correlates well with major histological, physiological and biochemical parameters, such as tissue colour, fibre diameter, contraction time, power output, endurance type, aerobic versus anaerobic activities, degree of resistance to fatigue and capillary density, as well as glycolytic versus oxidative metabolism. During fibre type transitions [123], triggered by graded exercise or chronic neuromuscular stimulation, myosin subunits show characteristic shifts in density and isoform expression patterns [36,124,125]. Detailed adaptations of the acto-myosin apparatus have been examined in a large number of gel-based proteomic studies, which have employed different types of physical activity (interval training, endurance exercise, vibration exercise during long-term bed rest, repeated eccentric exercise, downhill running, bouts of exhaustive exercise) in skeletal muscles from human, pig, rat and mouse [39,126,127,128,129,130,131,132,133,134,135,136,137,138,139,140], as well as chronic low-frequency electro-stimulated rabbit muscle [141,142].

Fast-to-slow transitions are clearly associated with the up-regulation of the slow/cardiac isoform MyHC-I and MLC1s, and a concomitant decrease in MLC1f, MLC2f and MLC3f [141] showing the usefulness of myosin changes for determining molecular adaptations during skeletal muscle transformation [11,36]. The specific effects of myostatin-related hypertrophy [143,144], hypoxia-induced skeletal muscle adaptations [145,146,147,148] and disuse-associated muscular atrophy (due to immobilization, denervation or neuromuscular unloading) [149,150,151,152,153,154,155,156,157,158] could also confirm distinct shifts in sarcomeric proteins due to fibre type transitions. The 2DGE top-down proteomic analysis of skeletal muscle development [159,160,161,162,163] versus muscle aging [164,165,166,167,168,169,170,171,172,173] has revealed interesting alterations in key proteins of the contractile apparatus [174,175]. For example, the gel-based analysis of fast-to-slow fibre transitions during muscle aging demonstrated a drastic increase of slow myosin light chain MLC2 in senescent skeletal muscle tissue [170], establishing this myosin subunit as an excellent marker of age-related fibre type shifting [176]. PTMs, such as altered phosphorylation levels of MLCs, appear to play a crucial role during skeletal muscle adaptations and fibre aging [177,178].

A key 2DGE technique is represented by fluorescence difference in-gel electrophoresis (DIGE) [179,180,181]. Figure 5 compares minimal protein labelling on lysine residues versus saturation protein labelling on cysteine residues prior to gel electrophoretic separation, and also illustrates the different CyDye labeling protocols used in fluorescent two-dye versus three-dye systems [182,183,184].

The 2D-DIGE method was originally developed by Minden and co-workers [185,186,187] and has been frequently modified as an efficient protein biochemical tool for studying large-scale changes in protein expression and interaction patterns [182,188,189,190]. The application of the 2D-DIGE technique for the comparative analysis of the skeletal muscle proteome from human and animal specimens has been described in detail in various methods papers [191,192,193,194,195]. For the optimum pre-electrophoretic DIGE labelling of protein fractions from tissue extracts, 50 μg protein aliquots can be differentially labelled with Cy2, Cy3 or Cy5 dyes and then separated on the same 2D gel, which eliminates gel-to-gel variations [181]. The development of sophisticated DIGE analysis software programs and improved protein identification approaches has greatly increased the quality of quantitative assessments of multiple proteomes on the same 2D gel [196,197,198].

An illustrative example of analysing changes in the skeletal muscle proteome by 2D-DIGE methodology is presented in Figure 6. Shown are 2D gel images from an optimization study of human *gastrocnemius* muscle. The figure clearly demonstrates the differing expression levels of distinct 2D protein spots in the gel electrophoretically separated Cy3-labelled controls (Sample A) versus physiologically challenged and Cy5-labelled specimens (Sample B). The protein clusters in the marked 2D-DIGE Zones 1 to 3 were densitometrically analysed and show unaltered proteoforms (Zone 1 and 2) versus drastically changed expression levels of distinct proteoforms (Zone 3).

### 3.3. Top-Down Proteomics of Troponins and Tropomyosins

As already described above for isoforms of myosins, the application of 2DGE top-down proteomics has also established regulatory proteins of the actin-containing thin filament as robust markers of fibre type shifting. Fast versus slow fibre specification in *gastrocnemius* and *soleus* muscle is clearly reflected by altered expression levels of the fast versus the slow isoform of tropomyosin-1-alpha [114]. During chronic electro-stimulation induced fast-to-slow muscle transitions, an increase in slow troponin TnT and TnI and concomitant decrease in fast troponin TnI were established by fluorescence 2D-DIGE analysis [142]. Troponins and tropomyosins are critical factors of the calcium-associated regulation of the excitation-contraction-relaxation cycle [199,200]. Tropomyosin interacts directly with actin filaments in an inhibitory role and is linked to troponin TnT, while inhibitory troponin TnI is regulated by the calcium-sensing TnC subunit [201].

In mature skeletal muscles, tropomyosin (TM) exists as slow and fast isoform combinations and troponin (TN) can be categorised as fast TnCf, slow TnCs, fast TnT1f to TnT4f, slow TnT1s, slow TnT2s, fast TnIf and slow TnIs [200]. These individual subunits relate in human muscle to slow TnT (P13805; *TNNT1* gene), fast TnT (P45378; *TNNT3* gene), slow TnI (P19237; *TNNI1* gene), fast TnI (P48788; *TNNI2* gene), slow TnC (P63316; *TNNC1* gene), fast TnC (P02585; *TNNC2* gene), TM-alpha-1 (P09493; *TPM1* gene), TM-alpha-3 (P06753; *TPM3* gene) and TM-alpha-4 (P67936; *TPM4* gene). The recent top-down proteomic analysis of tropomyosins in rat, pig and human muscle has established that tropomyosin isoforms Tpm1.1 and Tpm2.2 are the two major tropomyosin isoforms in swine and rat skeletal fibres [202]. In contrast, isoforms Tpm1.1, Tpm2.2, and Tpm3.12 were identified in human muscles. Since the age-related loss of skeletal muscle mass and contractile strength [203] plays an essential role in the frailty syndrome [204,205,206], top-down proteomics is instrumental for studying skeletal muscle aging. Gel-based proteomic surveys of senescent human skeletal muscles and established animal models of sarcopenia strongly suggest that a gradual shift to more oxidative metabolism and a slower mode of fibre contraction occurs during aging [176]. Besides myosin heavy and light chains, the regulatory elements tropomyosin and troponin also showed distinct fast-to-slow changes in proteoform expression levels [164,165,166,168,207]. Decreases in fast TnT and tropomyosin TM-alpha agree with the idea that age-associated abnormalities in the peripheral nervous system trigger denervation, faulty patterns of re-innervation and excitation-contraction-uncoupling, which indirectly affects the skeletal muscle phenotype and causes a gradual transformation to a slower-contracting muscle [176]. The phosphorylation status of the sarcomeric Z-disk component cypher, as well as PTMs in other sarcomeric proteins, was determined to be altered in aged muscle by quantitative top-down proteomics [208].

### 3.4. Analysis of Post-Translational Modifications Using Two-Dimensional Gel Electrophoresis

There are many PTMs including phosphorylation, glycosylation, acetylation and ubiquitination that are involved in a large variety of processes such as signal transduction pathways, protein trafficking, enzymatic activity or cell motility to name but a few [209]. Determining the precise role of PTMs in physiology and pathophysiology can decisively increase our biochemical understanding and help to take the correct approach when trying to modulate these modifications with respect to therapeutic actions. 2DGE is a highly effective top-down approach to resolve proteoforms, including post-translationally modified proteins. Thousands of proteins can be identified using optimized 2DGE when combined with advanced MS, and potentially many PTMs can be detected. Specific stains have been developed for phosphoproteins (Pro-Q Diamond) and glycoproteins (Pro-Q Emerald) to study specific PTMs using fluorescence 2DGE [210,211,212,213,214]. By using a combination of background-adjusted Pro-Q Diamond staining/total protein ratio values, differential phosphorylation for specific proteins can be detected. Phosphorylated proteins of interest can subsequently be digested, phospho-peptides enriched using titanium dioxide (TiO_2_) chromatography and specific phosphorylation sites identified by MS analysis. The advantages of separating protein by 2DGE and then identifying augmented phosphorylation signals means that researchers can focus on individual proteins and characterise their PTM status specifically, as opposed to high-throughput LC-MS/MS analysis of enriched peptides that represents a more global approach.

Reversible protein phosphorylation, predominantly involving serine, threonine or tyrosine residues, is one of the most important and intensely investigated PTMs [215]. As an established model system of sarcopenia of old age, 30-month old rat *gastrocnemius* muscle fibres were evaluated using phospho-specific Pro-Q Diamond staining of 2D gels [216]. Increased phosphorylation levels were shown for various contractile proteins including slow myosin light chain MLC2, tropomyosin alpha and actin, providing evidence that age-related muscle wasting has a complex pathology that is associated with significant changes in the abundance of phosphorylated proteins [177,178,208]. The importance of contractile protein phosphorylation was confirmed by a study on myosin binding protein MyBP-C. Verduyn and co-workers [217] induced differences in the degree of phosphorylation of the myofilaments to explore the role of phosphorylation of the cMyBP-C isoform in isolated rat ventricular myocytes under different physiological conditions, namely phosphorylation of myosin binding protein in quiescent hearts perfused with lidocaine at a relatively low calcium concentration, higher levels of phosphorylation obtained by β-adrenergic stimulation, enhancement of calmodulin-dependent protein kinase CaMK II activity by an increase in the time averaged intracellular calcium concentration, and/or by blockade of phosphatase activity. Using 2DGE and ProQ Diamond staining procedures, the endogenous phosphorylation levels of contractile proteins was determined for troponin cTnT, myosin light chain MLC1 and myosin light chain MLC2. Phosphorylation levels of the contractile proteins varied considerably with the experimental conditions used, with an approximate 2-fold increase in MLC2 phosphorylation in perfused contractile tissues stimulated at 5 Hz. Addition of the beta-agonist isoprenaline to the stimulated hearts resulted in 3.7-fold increases in both cMyBP-C and troponin cTnI phosphorylation [217].

Protein acetylation is involved in the regulation of protein stability and function during various cellular and physiological processes [218,219], including skeletal muscle atrophy [220] and age-related fibre wasting [221]. Acetylation of lysine residues on both histones and non-histone proteins is a reversible, dynamic protein modification regulated by lysine acetyltransferases and deacetylases, controlling a range of different diverse biological functions including protein-protein interactions, protein-DNA interactions, enzymatic activity and subcellular localization. High levels of dihydrolipoyllysine-residue acetyltransferase were detected in the nucleus in aged rat *triceps* muscle by fluorescence 2D-DIGE analysis and shown to be associated with increased protein acetylation [221]. Acetylation of lysine residues removes the positive charge of the side chain and therefore directly impacts the electrostatic status of the modified protein, allowing for separation associated with isoelectric focusing. Protein p*I*-shifts due to PTMs revealed that the theoretical p*I*-value of the non-acetylated form of tropomyosin-beta is close to the p*I*-value of its experimentally acetylated counterpart, while additional acetylation of tropomyosin (LEKTIDDLEETLASAK + acetyl (K); acetyl (N-term)) does not significantly affect the p*I*-value [222]. Interestingly, a variety of contractile proteins are changed in chronic Chagas disease, an often fatal outcome of *Trypanosoma cruzi* infection, which is characterized by severe cardiomyopathy and chronic skeletal muscle myositis and vasculitis [223,224]. Protein changes were evaluated using 2DGE of specimens from end-stage chronic Chagas disease patients to gain insight into its pathophysiology [225]. Several gel spots with differing p*I*-values and comparable molecular mass were identified by peptide mass fingerprinting as proteoforms of key structural and contractile proteins, including several forms of actin (ACTA1, ACTA2, ACTC, ACTG2, ACTN2), desmin, myosin (MYL3 and MYL7) and vimentin. The variations in their position within 2D slab gels could be a result of PTMs, such as acetylation or other processes.

Glycosylation is one of the most complex PTMs and is involved in many biological mechanisms including cell differentiation, intracellular signalling and protein trafficking [226]. Cieniewski-Bernard and co-workers [227] demonstrated the utility of 2DGE for the analysis of O-linked N-acetylglucosaminylation (O-GlcNAc) in skeletal muscle proteins, focusing on rat *gastrocnemius* muscle that is composed of both fast- and slow-contracting fibres. O-GlcNAc is an abundant and reversible type of glycosylation and especially found within the cytosolic and the nuclear compartments. The total level of O-GlcNAc proteins in rat after hind-limb unloading, a model of skeletal muscle atrophy, were also evaluated. The O-GlcNAc modification of myosin was found to be significant and verified using immunoblot analysis with an anti-O-GlcNAc antibody. The authors suggested that O-GlcNAc could be involved in the regulation of the polymerization of myosin in the thick filament assembly. The variation in the O-GlcNAc level measured after skeletal muscle unloading suggests a role for dynamic glycosylation patterns in muscle plasticity [227].

Ubiquitination is a key type of reversible PTM in proteins and plays a significant role in the regulation of many biological processes, including protein degradation and signal transduction [228]. Ubiquitin, a highly conserved 76-amino acid protein, can be added to substrate protein as a protein tag by the sequential actions of ubiquitin activating enzyme (E1), ubiquitin conjugating enzyme (E2) and ubiquitin protein ligase (E3). Using fractions of *gastrocnemius* muscles isolated from Trim32 knockout mice (T32KO) compared to wild type, 36 proteins with altered abundance in T32KO muscle were identified using a combination of fluorescence 2D-DIGE and LC-MS/MS analysis [229]. Mutations in the *TRIM32* gene, which encodes an E3 ubiquitin ligase containing the tripartite motif, have been implicated in the pathogenesis of limb girdle muscular dystrophies. Of the significant proteins identified using this model system, the fast muscle isoform of troponin TnT was shown to decrease in abundance in T32KO muscle, a result that supports selective type II (fast) fibre atrophy in T32KO mice.

## 4. Conclusions

MS-based proteomics has increasingly become the most used method for analysing complex protein populations in biological samples, such as skeletal muscle tissues, facilitated by rapidly advancing hardware and software products. Bioanalytical MS analysis can be employed for experiments focusing on muscle protein profiling, protein quantification, analysis of muscle protein interactions and studying PTMs. Within the field of skeletal muscle biochemistry, there is still a position in proteomics for 2D-gels, both in terms of a standalone technology and as a complementary approach to MS analysis. When used in combination with narrow range pH-gels, 2DGE can resolve more than 5000 distinct protein spots, supplying the end-user with a significant amount of data to interrogate specific biological features of interest, such as muscle fibre type specification or skeletal muscle transitions. The ease of the quantitative analysis of intact muscle proteins using approaches like fluorescence 2D-DIGE makes the comparison of normal versus adapting tissues or healthy versus diseased skeletal muscle samples very achievable. As 2D-gel patterns provide detailed information regarding the position of protein spots on large slab gels (isoelectric point and molecular mass), identifying PTMs that affect their charge and molecular mass is a particular niche application for 2DGE. Compatibility with a variety of proteomic workflows for the systematic identification of skeletal muscle proteins and to characterize their amino acid sequences, and using antibody-based analysis before electrophoresis (such as immuno-affinity purification) or after electrophoretic separation (such as immunoblotting), will ensure that 2DGE remains a relevant laboratory techniques for the next generation of muscle researchers.

## Figures and Tables

**Figure 1 proteomes-07-00025-f001:**
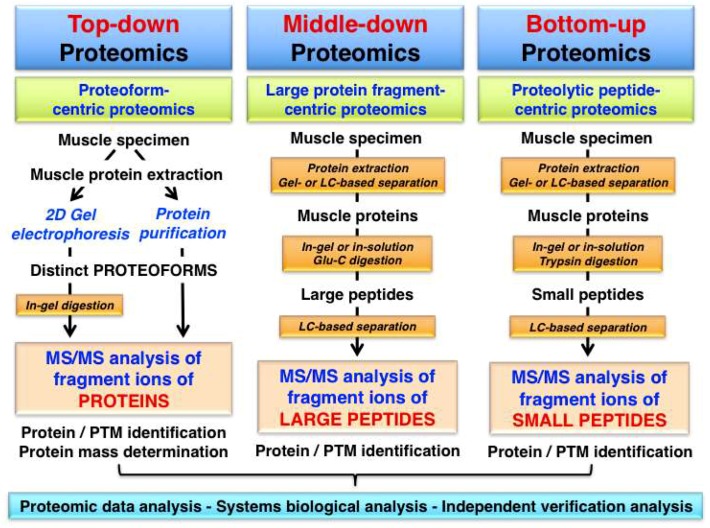
Overview of proteomic approaches used for the identification and characterization of skeletal muscle proteins and their post-translational modifications (PTMs). The diagram shows differences in the bioanalytical workflow involved in top-down proteomics, middle-down proteomics and bottom-up proteomics, which is characterized by the mass spectrometry (MS) analysis of intact proteins/proteoforms, protein fragments and peptides, respectively.

**Figure 2 proteomes-07-00025-f002:**
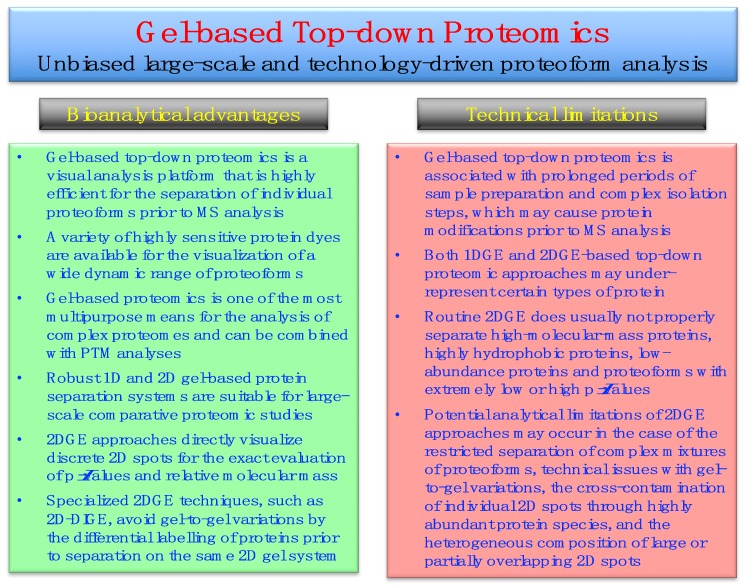
Overview of the bioanalytical advantages versus potential technical limitations of gel-based top-down proteomics, especially in relation to two-dimensional gel electrophoresis (2DGE) approaches.

**Figure 3 proteomes-07-00025-f003:**
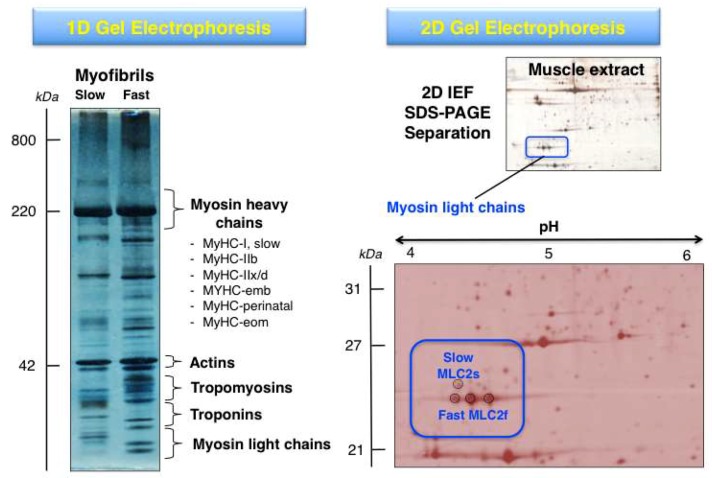
Comparison of gel electrophoretic images using 1D (one-dimensional) GE versus 2D (two-dimensional) GE analysis of contractile proteins from rodent skeletal muscle. The 1D gel shows the separation of isolated myofibrils from slow versus fast muscles and the relative position of key contractile proteins from the thick and thin filaments. The MS analysis of 1D gel zones has established the presence of isoforms of myosin heavy chain, myosin light chain, actin, troponin and tropomyosin. The 2DGE image of a separated skeletal muscle extract and its enlargement highlights the MS-based identification of proteoforms of myosin light chain (slow MLC2/P51667 and fast MLC2/P97457).

**Figure 4 proteomes-07-00025-f004:**
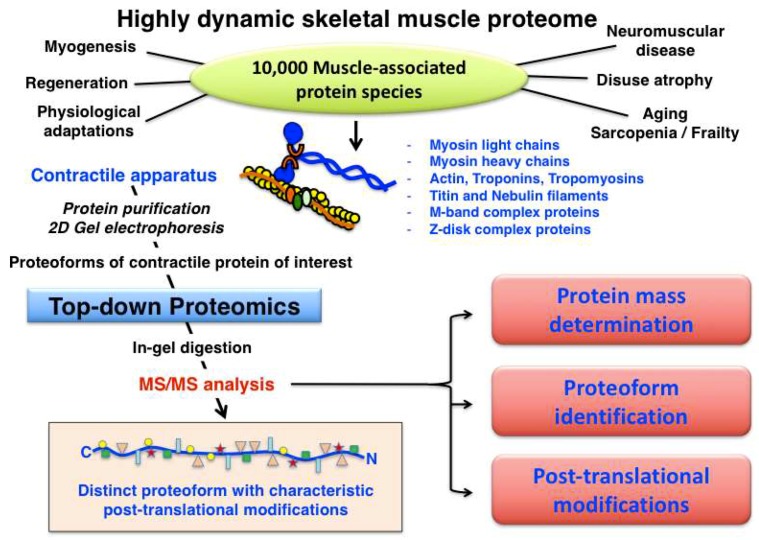
Overview of top-down proteomic analysis of contractile proteins from skeletal muscle.

**Figure 5 proteomes-07-00025-f005:**
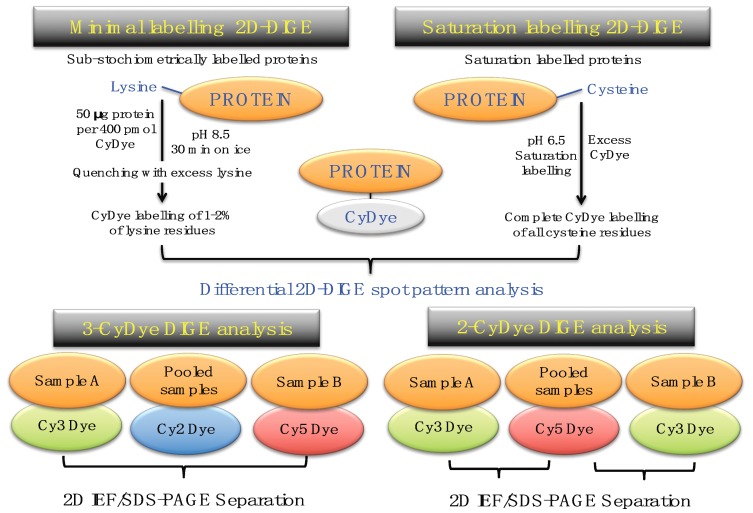
Overview of fluorescence two-dimensional difference in-gel electrophoresis (2D-DIGE) techniques for the comparative analysis of proteoforms. The upper panel shows the two main DIGE approaches for the systematic pre-electrophoretic labelling of protein fractions using fluorescent CyDyes. These methods utilise minimal sub-stochiometric labelling of assessable lysines in proteins or saturation labelling of assessable cysteines in proteins. The lower panel outlines three-dye (Cy2, Cy3, Cy5) versus two-dye (Cy3, Cy5) DIGE labelling methodology.

**Figure 6 proteomes-07-00025-f006:**
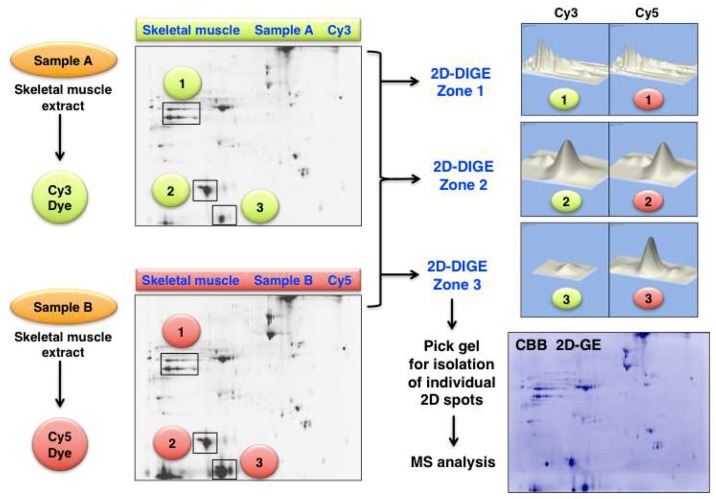
Fluorescence two-dimensional difference in-gel electrophoresis (2D-DIGE) of human *gastrocnemius* muscle. The 2DGE images on the left show Cy3 (Sample A) versus Cy5 (Sample B) labelled and separated muscle proteins. The densitometric analysis of the marked 2D-DIGE Zones 1 to 3 is shown on the right and depicts clusters of mostly unaltered proteins (Zone 1 and 2) versus significantly altered expression levels (Zone 3). The routine identification of distinct proteoforms is performed by MS analysis of individual 2D spots from Coomassie Brilliant Blue (CBB) stained pick gels.

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
