# Peer review of "Characterization of Contractile Proteins from Skeletal Muscle Using Gel-Based Top-Down Proteomics"

_proteomes, 2019, doi:10.3390/proteomes7020025_

Round 1

Reviewer 1 Report

The manuscript by Dowling et al reviews literature regarding the use of two-dimensional gel electrophoresis (2DGE), a top-down proteomic approach, to investigate contractile proteins in skeletal muscle. The review provides evidence that these techniques provide data to improve the biochemical understanding of how individual proteins and proteoforms contribute to the structure and function of skeletal muscle. This article benefits the current field by providing an up to date review of work implementing 2DGE techniques. I have no major concerns with the work, but the manuscript could be improved by considering the following comments

The manuscript is devoid of 2DGE images. The strength of a 2DGE is that it is a visual platform where individual proteoforms can be separated prior to MS analysis. Connected with this point, I invitethe authors comment on 2D gel databases, many of which are no longer actively maintained. Is this a problem to the further development of the field/ sharing of information?

There is a lack of emphasis on the need to identify site-specific modifications. For example, Pro-Q staining cannot distinguish single from multiple phosphorylations. Also, it may be extremely challenging to map all modifications that determine a particular gel spot/ proteoform, e.g. could be a composite of several phosphorylation, acetylation, ubiquitination, etc. events that are unique to a proteoform. This is a missed opportunity to highlight the problems facing high-throughput LC-MS/MS analysis of enriched peptides.

The manuscript includes an overview of mass-spectrometry in proteomics (lines 33-46). This might be introduced better by beginning with information about 2DGE separation (similar to that in section 3.5) and how mass spectrometry can be combined with protein-level separation to generate insight to the muscle proteome. The rationale for a top-down rather than bottom-up approach could be justified in more strongly/ in more detail (Figure 1 could be amended to highlight this). For example, the authors describe the structure and plasticity of muscle (lines 72-87) but miss an opportunity to report the full complexity of the myofibrillar proteome i.e. differences caused not just by changes in protein expression, but through numerous different isoforms/splice variants/ proteoforms – the strength of 2DGE is that this level of complexity can be investigated, whereas bottom-up peptide analysis will be blind to some of these features.

I encourage the authors to explore the physiological meaning to the literature findings in more detail, e.g. in section 3.4 (lines 227-236) it is not clear what the role of the troponin and tropomyosin isoforms are in the aging process. 

Section 3.2 discusses myosin heavy chain isoforms. Unless I am mistaken, these are not readily resolved by standard 2DGE. If I am wrong, is it possible to show 2DGE separation of MyHC isoforms? 

The manuscript does not include information on the repeatability or reproducibility of 2DGE and staining techniques, or how these compare to bottom-up LC-MS/MS approaches.

Traditional protein names and gene ID’s are used. Given this is a proteomics paper (that focuses on proteoforms), it would be better to use UniProt ID’s that also detail isoforms, splice variations and post-translational modifications, rather than gene names.

The abstract uses the term “species” but the main text uses the newer/more appropriate term “proteoforms”. 

Author Response

Response to comments by Reviewer 1:

We would like to thank Reviewer 1 for the constructive criticism of our review article. In conjunction with the comments by the other 3 reviewers, we have revised our manuscript accordingly. This included the addition of 4 new figures with representative gel images, including the 2D-DIGE technique, and additional discussions of technical and biological issues of gel-based proteomics, as well as more detailed descriptions of certain aspects of skeletal muscle proteomics.

Please find below point-by-point responses to individual comments by Reviewer 1:

Comment 1: ‘The manuscript by Dowling et al reviews literature regarding the use of two-dimensional gel electrophoresis (2DGE), a top-down proteomic approach, to investigate contractile proteins in skeletal muscle. The review provides evidence that these techniques provide data to improve the biochemical understanding of how individual proteins and proteoforms contribute to the structure and function of skeletal muscle. This article benefits the current field by providing an up to date review of work implementing 2DGE techniques. I have no major concerns with the work, but the manuscript could be improved by considering the following comments: The manuscript is devoid of 2DGE images. The strength of a 2DGE is that it is a visual platform where individual proteoforms can be separated prior to MS analysis. Connected with this point, I invite the authors comment on 2D gel databases, many of which are no longer actively maintained. Is this a problem to the further development of the field/ sharing of information?’.

Response: To address these points, we have added new figures (Fig. 2, Fig. 3, Fig. 5 and Fig. 6; whereby old Fig. 2 is now re-numbered as Fig. 4) to the revised manuscript with representative images of gel-based analyses of skeletal muscle samples, as well as added comments and references on 2DGE databases.

New Figures on Pages 4, 6, 8 and 9:

Figure 2. Overview of the bioanalytical advantages versus potential technical limitations of gel-based top-down proteomics, especially in relation to 2DGE approaches.

Figure 3. Comparison of gel electrophoretic images using 1DGE versus 2DGE analysis of contractile proteins from rodent skeletal muscle. The 1D gel shows the separation of isolated myofibrils from slow versus fast muscles and the relative position of key contractile proteins from the thick and thin filaments. The MS analysis of 1D gel zones has established the presence of isoforms of myosin heavy chain, myosin light chain, actin, troponin and tropomyosins. The 2DGE image of a separated skeletal muscle extract and its enlargement highlights the MS-based identification of proteoforms of myosin light chain (slow MLC2/P51667 and fast MLC2/P97457).

Figure 5. Overview of fluorescence two-dimensional difference in-gel electrophoresis (2D-DIGE) techniques for the comparative analysis of proteoforms. The upper panel shows the two main DIGE approaches for the systematic pre-electrophoretic labelling of protein fractions using fluorescent CyDyes. These methods utilize minimal sub-stochiometric labelling of assessable lysines in proteins or saturation labelling of assessable cysteines in proteins. The lower panel outlines 3-dye (Cy2, Cy3, Cy5) versus 2-dye (Cy3, Cy5) DIGE labelling methodology.

Figure 6. Fluorescence two-dimensional difference in-gel electrophoresis (2D-DIGE) of human gastrocnemius muscle. The 2DGE images on the left show Cy3 (Sample A) versus Cy4 (Sample B) labelled and separated muscle proteins. The densitometric analysis of the marked 2D-DIGE Zones 1 to 3 is shown on the right and depicts clusters of mostly unaltered proteins (Zone 1 and 2) versus significantly altered expression levels (Zone 3). The routine identification of distinct proteoforms is performed by MS analysis of individual 2D spots from Coomassie Brilliant Blue (CBB) stained pick gels.

Revised text in relation to the description of new figures:

Revised Pages 4-5: ‘Bioanalytical advantages versus potential technical limitations of gel-based top-down proteomics, especially those issues associated with 2DGE approaches [42], are listed in Figure 2.… Gel-based methods are robust, highly reproducible and reliable protein separation techniques that are ideally suited for large-scale surveys of proteoforms. Staining of gel electrophoretically separated proteins can be achieved with a wide range of highly sensitive dyes, such as colloidal Coomassie Brilliant Blue, silver stains and various fluorescent dyes, as extensively reviewed [75-80]. The analytical strength of gel-based methods, as compared to protein separation by liquid chromatography, is the direct visualization of proteins within 1D gel bands or 2D gel spots. Thus, the position of discrete protein spots in 2D gels provides a swift and reliable way for determining the characteristic combination of the relative molecular mass and the pI-value of a specific proteoform of interest’.

Revised Page 6: ‘Figure 3 shows representative 1DGE and 2DGE images of separated muscle protein populations. These approaches are routinely used for the proteomic identification of distinct proteoforms of contractile components’.

Revised Pages 8-9: ‘A key 2DGE technique is represented by fluorescence difference in-gel electrophoresis (DIGE) [179-181]. Figure 5 compares minimal protein labelling on lysine residues versus saturation protein labelling on cysteine residues prior to gel electrophoretic separation, and also illustrates the different CyDye labeling protocols used in fluorescent 2-dye versus 3-dye systems [182-184]. … The 2D-DIGE method was originally developed by Minden and co-workers [185-187] and has been frequently modified as an efficient protein biochemical tool for studying large-scale changes in protein expression and interaction patterns [182, 188-190]. The application of the 2D-DIGE technique for the comparative analysis of the skeletal muscle proteome from human and animal specimens has been described in detail in various methods papers [191-195]. For the optimum pre-electrophoretic DIGE labelling of protein fractions from tissue extracts, 50μg protein aliquots can be differentially labeled with Cy2, Cy3 or Cy5 dyes and then separated on the same 2D gel, which eliminates gel-to-gel variations [181]. The development of sophisticated DIGE analysis software programs and improved protein identification approaches has greatly increased the quality of quantitative assessments of multiple proteomes on the same 2D gel [196-198]. … An illustrative example of analysing changes in the skeletal muscle proteome by 2D-DIGE methodology is presented in Figure 5. Shown are 2D gel images from an optimization study of human gastrocnemius muscle. The figure clearly demonstrates the differing expression levels of distinct 2D protein spots in the gel electrophoretically separated Cy3-labelled controls (Sample A) versus physiologically challenged and Cy5-labelled specimens (Sample B). The protein clusters in the marked 2D-DIGE Zones 1 to 3 were densitometrically analysed and show unaltered proteoforms (Zone 1 and 2) versus drastically changed expression levels of distinct proteoforms (Zone 3).

The new References [179] to [198] on the 2D-DIGE method have been added to the revised list of references.

Revised text on Page 2 on 2DGE databases: ‘Since 2DGE approaches present a visual platform for the efficient separation of individual proteoforms prior to MS analysis, the availability of internet-based comparisons via international 2D-gel databases is a strength of gel-based proteomics. A considerable number of 2DGE databases were established, especially in the period 1993-2008 [16-28], and include SWISS-2DPAGE, World-2DPAGE, Flicker, UAB Proteomics Database, AGML Central, GELBANK, WEB P.A.G.E, Open2Dprot Project, LECB 2-D PAGE Gel Images Data Sets, ProteomeWeb, PHProteomicsDB, pProRep and DynaProt 2D. These extensive collections of 2DGE images and bioinformatic analysis tools can be highly useful for the comparative analysis of newly established 2DGE images, the systematic cataloguing of proteoforms using 2DGE techniques and the integration of gel-based proteomic findings and mass spectral data [28]. Comparative image analyses can also be helpful for the swift identification of 1D protein bands and 2D protein spots within archived gels that were produced prior to the establishment of MS-based proteomics [29, 30]. Unfortunately, many of these 2DGE image databases are no longer actively maintained, which constitutes a drawback in the further development of the field of gel-based proteomics and hampers the efficient sharing of proteomic data generated by 2DGE analyses. Hopefully, international efforts will be able to re-establish functional and internet-based databanks for future gel electrophoretic studies and the efficient comparison of proteomic maps’.

The new References [16] to [28] on the 2DGE databases have been added to the revised list of references.

Comment 2: ‘There is a lack of emphasis on the need to identify site-specific modifications. For example, Pro-Q staining cannot distinguish single from multiple phosphorylations. Also, it may be extremely challenging to map all modifications that determine a particular gel spot/ proteoform, e.g. could be a composite of several phosphorylation, acetylation, ubiquitination, etc. events that are unique to a proteoform. This is a missed opportunity to highlight the problems facing high-throughput LC-MS/MS analysis of enriched peptides’.

Response:

To address this point, we have added the following statement in section 3.4. on revised Page 10: ‘By using a combination of background-adjusted Pro-Q Diamond/total protein ratio values, differential phosphorylation for specific proteins can be detected. Phosphorylated proteins of interest can subsequently be digested, phospho-peptides enriched using titanium dioxide (TiO2) chromatography and specific phosphorylation sites identified by MS analysis. The advantages of separating protein by 2DGE and then identifying augmented phosphorylation signals means that researchers can focus on individual proteins and characterize their PTM status specifically, as opposed to high-throughput LC-MS/MS analysis of enriched peptides that represents a more global approach’.

Comment 3: ‘The manuscript includes an overview of mass-spectrometry in proteomics (lines 33-46). This might be introduced better by beginning with information about 2DGE separation (similar to that in section 3.5) and how mass spectrometry can be combined with protein-level separation to generate insight to the muscle proteome. The rationale for a top-down rather than bottom-up approach could be justified in more strongly/ in more detail (Figure 1 could be amended to highlight this). For example, the authors describe the structure and plasticity of muscle (lines 72-87) but miss an opportunity to report the full complexity of the myofibrillar proteome i.e. differences caused not just by changes in protein expression, but through numerous different isoforms/splice variants/ proteoforms – the strength of 2DGE is that this level of complexity can be investigated, whereas bottom-up peptide analysis will be blind to some of these features’.

Response: To address this point, we have moved the introductory statement from section 3.5 to the Introduction section. Following 2 introductory sentences on the general principles of proteomics, the revised introduction states now early on the importance of gel-based proteomics, including the quoting of relevant references on 2DGE. This introduction into gel-based proteomics is then directly followed by the already above described new section on 2DGE approaches as visual platforms for the efficient separation of individual proteoforms prior to MS analysis, as well as the availability of internet-based comparisons via international 2D-gel databases. In sections that follow the introduction, details are given on proteoforms and their detection by gel-based proteomics. We have also added a new figure (Fig. 2) listing advantages versus potential limitations of gel-based top-down proteomics to address the reviewer’s concern.

Page 1 (revised first paragraph of Introduction section): ‘…within specific protein constituents [6-9]. Gel-based proteomics is one of the most multipurpose means for the analysis of complex proteomes, such as seen in skeletal muscle tissues [10-12]. Two-dimensional gel electrophoresis (2DGE) exemplifies an exceptionally robust strategy to separate, identify, and quantify proteins when used in combination with MS analysis or various immunological tests [13-15]. While 2DGE is an effective technique for the systematic analysis of complex protein lysates, the ability to directly visualise changes in the abundance of proteins/post-translational modifications (PTM) is a powerful application’

Figure 1 was revised and the section on top-down proteomics now lists the word ‘PROTEOFORMS’ to highlight the difference between the top-down versus bottom-up approaches.

The following text was added to Pages 4-5 in relation to new Figure 2: ‘… Bioanalytical advantages versus potential technical limitations of gel-based top-down proteomics, especially those issues associated with 2DGE approaches [42], are listed in Figure 2. … Gel-based methods are robust, highly reproducible and reliable protein separation techniques that are ideally suited for large-scale surveys of proteoforms. Staining of gel electrophoretically separated proteins can be achieved with a wide range of highly sensitive dyes, such as colloidal Coomassie Brilliant Blue, silver stains and various fluorescent dyes, as extensively reviewed [75-80]. The analytical strength of gel-based methods, as compared to protein separation by liquid chromatography, is the direct visualization of proteins within 1D gel bands or 2D gel spots. Thus, the position of discrete protein spots in 2D gels provides a swift and reliable way for determining the characteristic combination of the relative molecular mass and the pI-value of a specific proteoform of interest. …’.

The new References [75] to [80] on protein dyes used in 2DGE have been added to the revised list of references.

Comment 4: ‘I encourage the authors to explore the physiological meaning to the literature findings in more detail, e.g. in section 3.4 (lines 227-236) it is not clear what the role of the troponin and tropomyosin isoforms are in the aging process’. 

Response: To address this point, we have added text on troponin and tropopmysoin during aging on revised Pages 10 as follows: ‘…is instrumental for studying skeletal muscle aging. Gel-based proteomic surveys of senescent human skeletal muscles and established animal models of sarcopenia strongly suggest that a gradual shift to more oxidative metabolism and a slower mode of fibre contraction occurs during aging [176]. Besides myosin heavy and light chains, the regulatory elements tropomyosin and troponin also showed distinct fast-to-slow changes in proteoform expression levels [164-166, 168, 207]. Decreases in fast TnT and tropomyosin TM-alpha agree with the idea that age-associated abnormalities in the peripheral nervous system trigger denervation, faulty patterns of re-innervation and excitation-contraction-uncoupling, which indirectly affects the muscle phenotype and causes a gradual transformation to a slower-contracting mechanism [176]. The phosphorylation status of the sarcomeric Z-disk component cypher, as well as post-translational modifications in other sarcomeric proteins, was determined to be altered in aged muscle by quantitative top-down proteomics [208].

New reference [207]: Donoghue, P., Staunton, L., Mullen, E., Manning, G., and Ohlendieck K. (2010) DIGE analysis of rat skeletal muscle proteins using nonionic detergent phase extraction of young adult versus aged gastrocnemius tissue. J. Proteomics 73, 1441-1453.

Comment 5: ‘Section 3.2 discusses myosin heavy chain isoforms. Unless I am mistaken, these are not readily resolved by standard 2DGE. If I am wrong, is it possible to show 2DGE separation of MyHC isoforms?’.

Response: The reviewer is correct. The routine biochemical analysis of MyHC isoform expression patterns in muscle preparations is usually carried out by 1D gel electrophoresis approaches that use optimised running conditions for the efficient separation of fast versus slow MyHCs. This analysis can also be combined with immunoblotting, peptide sequencing and/or mass spectrometry. The details on MyHCs are discussed in this section, because they are critical for fibre type specification and our understanding of skeletal muscle plasticity at the level of contractile kinetics and sarcomeric organization.

Since the other reviewers also suggested the introduction of illustrative gel images, we have added a new Figure 3. Our laboratory has recently used the GeLC-MS/MS method for the proteomic survey of myofibrils.A large number of distinct myosin heavy chains, i.e. slow MyHC-I, fast MyHC-IIb, fast MyHC-IIx/d, MYHC-EMB, perinatal MyHC and MyHC-EOM, and myosin light chains MCL1/3, slow MLC2, fast MLC2 and MLC3 were identified, demonstrating an excellent degree of sensitivity for protein identification by GeLC-MS/MS. In addition, both the muscle and cardiac isoform of α-actin, as well as α- and β-tropomyosins and the troponin subunits TnI, TnC and TnT were identified in this study. A similar 1D gel and identified gel zones are shown in new Figure 3.

New Figure 3. Comparison of gel electrophoretic images using 1DGE versus 2DGE analysis of contractile proteins from rodent skeletal muscle. The 1D gel shows the separation of isolated myofibrils from slow versus fast muscles and the relative position of key contractile proteins from the thick and thin filaments. The MS analysis of 1D gel zones has established the presence of isoforms of myosin heavy chain, myosin light chain, actin, troponin and tropomyosins. The 2DGE image of a separated skeletal muscle extract and its enlargement highlights the MS-based identification of proteoforms of myosin light chain (MLC2s and MLC2f).

Comment 6: ‘The manuscript does not include information on the repeatability or reproducibility of 2DGE and staining techniques, or how these compare to bottom-up LC-MS/MS approaches’.

Response: To address this point, we have added a new Figure 2 that lists the advantages of gel-based top-down proteomics and have added a description of the robust and highly reproducible nature of gel electrophoresis on revised Pages 4-5, as well as mentioned the main staining techniques and refer to extensive reviews of this topic, as follows: ‘Gel-based methods are robust, highly reproducible and reliable protein separation techniques that are ideally suited for large-scale surveys of proteoforms. Staining of gel electrophoretically separated proteins can be achieved with a wide range of highly sensitive dyes, such as colloidal Coomassie Brilliant Blue, silver stains and various fluorescent dyes, as extensively reviewed [75-80]. The analytical strength of gel-based methods, as compared to protein separation by liquid chromatography, is the direct visualization of proteins within 1D gel bands or 2D gel spots. Thus, the position of discrete protein spots in 2D gels provides a swift and reliable way for determining the characteristic combination of the relative molecular mass and the pI-value of a specific proteoform of interest.’.

In the new section on 2D-DIGE (new Figures 5 and 6), we have also quoted a variety of new references that have demonstrated the reproducibility of this representative 2DGE method.

Comment 7: ‘Traditional protein names and gene ID’s are used. Given this is a proteomics paper (that focuses on proteoforms), it would be better to use UniProt ID’s that also detail isoforms, splice variations and post-translational modifications, rather than gene names’.

Response: To address this point, we have revised the manuscript text accordingly and added Uniprot IDs when general descriptions focus on specific proteoforms, such as the introduction into the various myosin heavy chains, myosin light chains, tropomyosins and troponins.

Comment 8: ‘The abstract uses the term “species” but the main text uses the newer/more appropriate term “proteoforms”.’.

Response: We agree and have removed the word ‘species’ from the Abstract section. 

Revised Abstract: The mass spectrometric analysis of skeletal muscle proteins has used both peptide-centric and protein-focused approaches. The term ‘top-down proteomics’ is often used in relation to studying purified proteoforms and their post-translational modifications. Two-dimensional gel electrophoresis, in combination with peptide generation for the identification and characterization of intact proteoforms being present in 2D spots, plays a critical role in specific applications of top-down proteomics. A decisive bioanalytical advantage of gel-based and top-down approaches is the initial bioanalytical focus on intact proteins, which usually enables the swift identification and detailed characterisation of specific proteoforms. In this review, we describe the usage of two-dimensional gel electrophoretic top-down proteomics and related approaches for the systematic analysis of key components of the contractile apparatus, with a special focus on myosin heavy and light chains and their associated regulatory proteins. The detailed biochemical analysis of proteins belonging to the thick and thin skeletal muscle filaments has decisively improved our biochemical understanding of structure-function relationships within the contractile apparatus. Gel-based and top-down proteomics has clearly established a variety of slow and fast isoforms of myosin, troponin and tropomyosin as excellent markers of fibre type specification and dynamic muscle transition processes.

Reviewer 2 Report

The manuscript “Characterization of contractile proteins from skeletal muscle using two-dimensional gel electrophoretic  top-down proteomics”  is well written and will be useful for people working in area of muscle proteomics. But there are some remarks.

1)     The title. As not only 2DE is used for analysis of contractile proteins, the title can be  “Characterization of contractile proteins from skeletal muscle using gel-based  top-down proteomics.

2)     It seems that the manuscript can be improved by adding the specific graphics. For instance 2DE of contractile proteins, proteins of fast and slow muscles (1DE, 2DE)…

3)     The terms “Two dimensional gel electrophoresis” and “postranslation modifications” are used very often, so it’s reasonable to use  abbreviations -  2DE, PTM

Author Response

Response to comments by Reviewer 2:

We would like to thank Reviewer 2 for the constructive criticism of our review article. In conjunction with the comments by the other 3 reviewers, we have revised on our manuscript accordingly. This included the addition of new figures with representative gel images, including the 2D-DIGE technique, and additional discussions of technical and biological issues of gel-based proteomics, as well as more detailed descriptions of certain aspects of skeletal muscle proteomics.

Please find below point-by-point responses to individual comments by Reviewer 2:

Comment 1: ‘The manuscript “Characterization of contractile proteins from skeletal muscle using two-dimensional gel electrophoretic top-down proteomics” is well written and will be useful for people working in area of muscle proteomics. But there are some remarks. 1) The title. As not only 2DE is used for analysis of contractile proteins, the title can be “Characterization of contractile proteins from skeletal muscle using gel-based top-down proteomics.

Response: We agree and have changed the title accordingly to ‘Characterization of contractile proteins from skeletal muscle using gel-based top-down proteomics’.

Comment 2: ‘2) It seems that the manuscript can be improved by adding the specific graphics. For instance 2DE of contractile proteins, proteins of fast and slow muscles (1DE, 2DE)…’.

Response: We agree and have added 4 new figures of which 3 figures represent illustrative overviews (Fig. 3, Fig. 5 and Fig. 6) in the revised manuscript with representative images of gel-based analyses of skeletal muscle samples. This includes 1DGE images of fast versus slow myofibrils, 2DGE images of muscle extracts, highlighted 2DGE images of MLC proteoforms and comparative fluorescence 2D-DIGE images.

New Figures:

Figure 3. Comparison of gel electrophoretic images using 1DGE versus 2DGE analysis of contractile proteins from rodent skeletal muscle. The 1D gel shows the separation of isolated myofibrils from slow versus fast muscles and the relative position of key contractile proteins from the thick and thin filaments. The MS analysis of 1D gel zones has established the presence of isoforms of myosin heavy chain, myosin light chain, actin, troponin and tropomyosins. The 2DGE image of a separated skeletal muscle extract and its enlargement highlights the MS-based identification of proteoforms of myosin light chain (slow MLC2/P51667 and fast MLC2/P97457).

Figure 5. Overview of fluorescence two-dimensional difference in-gel electrophoresis (2D-DIGE) techniques for the comparative analysis of proteoforms. The upper panel shows the two main DIGE approaches for the systematic pre-electrophoretic labelling of protein fractions using fluorescent CyDyes. These methods utilize minimal sub-stochiometric labelling of assessable lysines in proteins or saturation labelling of assessable cysteines in proteins. The lower panel outlines 3-dye (Cy2, Cy3, Cy5) versus 2-dye (Cy3, Cy5) DIGE labelling methodology.

Figure 6. Fluorescence two-dimensional difference in-gel electrophoresis (2D-DIGE) of human gastrocnemius muscle. The 2DGE images on the left show Cy3 (Sample A) versus Cy4 (Sample B) labelled and separated muscle proteins. The densitometric analysis of the marked 2D-DIGE Zones 1 to 3 is shown on the right and depicts clusters of mostly unaltered proteins (Zone 1 and 2) versus significantly altered expression levels (Zone 3). The routine identification of distinct proteoforms is performed by MS analysis of individual 2D spots from Coomassie Brilliant Blue (CBB) stained pick gels.

Comment 3: ‘3) The terms “Two dimensional gel electrophoresis” and “postranslation modifications” are used very often, so it’s reasonable to use abbreviations -  2DE, PTM’.

Response: We agree and have added abbreviations of frequently used words, such as 2DGE and PTM.

Revised Pages 12-13:

Abbreviations

The following abbreviations are used in this manuscript:

1D: one-dimensional

2D: two-dimensional

DIGE: difference in-gel electrophoresis

GE: gel electrophoresis

IEF: isoelectric focusing

LC: liquid chromatography

MLC: myosin light chain 

MS: mass spectrometry

MyHC, myosin heavy chain

PAGE: polyacrylamide gel electrophoresis

pI: isoelectric point

PTM: post-translational modification

SDS: sodium dodecyl sulfate

TM: tropomyosin

TN: troponin

Reviewer 3 Report

This manuscript is a review on studies of isoforms and modified forms of skeletal muscle proteins by electrophoresis-based proteomics. Any tissue contains a lot of protiens that are present as severel different isoforms and posttranslationally modified forms. However, in bottom-up LC-MS/MS-based proteomics, most fragments from different forms of a protein are mixed together, and thus are unable to analyze separately. This review illuminate the merit of 2DE in analyzing protein isoforms/modified forms, and summarized the studies on skeletal muscle using 2DE. Therefore, in my opinion, this manuscript will be a valuable review when published, but there are several points to be considered. 

Specific points,

1) The term "Top-down proteomics" seems to me a little confusing. My understanding is that it originally represented MS/MS analysis of intact proteins without protease digestion. If proteins are analyzed after protease digestion, some other words such as "gel-based proteomics" might be more specific and easy-to-undesrstand. If the word "Top-down proteomics" will still be used, it would be necessary to make the meaning clearer. Although it was mentioned in Chapter 2, the difference between "Top-down proteomics" and "Bottom-up proteomics" should be explained more simply and clearly. 

2) In the procedure shown in Figure 2, there is a s "Top-down proteomics" step before "MS/MS analysis". However, it is not clear what this step is. Is MS/MS analysis performed after digestion? If so, isn't it just mean in-gel digestion? In addition, is it really possible to perform "Absolute mass determination" ? 

Minor points,

3) Including a 2D-gel image showing muscle protein heterogeneity in this review would be helpful to understand the usefulness of 2D-PAGE in the study of skeletal muscles. 

Author Response

Response to comments by Reviewer 3:

We would like to thank Reviewer 3 for the constructive criticism of our review article. In conjunction with the comments by the other 3 reviewers, we have revised on our manuscript accordingly. This included the addition of new figures with representative gel images, including the 2D-DIGE technique, and additional discussions of technical and biological issues of gel-based proteomics, as well as more detailed descriptions of certain aspects of skeletal muscle proteomics.

Please find below point-by-point responses to individual comments by Reviewer 3:

Comment 1: ‘This manuscript is a review on studies of isoforms and modified forms of skeletal muscle proteins by electrophoresis-based proteomics. Any tissue contains a lot of protiens that are present as severel different isoforms and posttranslationally modified forms. However, in bottom-up LC-MS/MS-based proteomics, most fragments from different forms of a protein are mixed together, and thus are unable to analyze separately. This review illuminate the merit of 2DE in analyzing protein isoforms/modified forms, and summarized the studies on skeletal muscle using 2DE. Therefore, in my opinion, this manuscript will be a valuable review when published, but there are several points to be considered. Specific points, 1) The term "Top-down proteomics" seems to me a little confusing. My understanding is that it originally represented MS/MS analysis of intact proteins without protease digestion. If proteins are analyzed after protease digestion, some other words such as "gel-based proteomics" might be more specific and easy-to-undesrstand. If the word "Top-down proteomics" will still be used, it would be necessary to make the meaning clearer. Although it was mentioned in Chapter 2, the difference between "Top-down proteomics" and "Bottom-up proteomics" should be explained more simply and clearly’.

Response: We like to thank Reviewer 3 for this critical comment and have addressed this issue in the revised version of our manuscript. In this review, we would like to keep the term ‘top-down proteomics’ in relation to the combination of 2DGE and MS analysis. Our preference is on using the definition of ‘top-down proteomics’ in a broader sense than just the focused MS/MS analysis of an intact protein. We would like to suggest that the term also includes approaches that start the proteomic analysis with a separated and thus isolated proteoform of a particular protein and then employ differing MS methods for protein/peptide identification or PTM analysis. We have addressed this the revised manuscript as follows:

Revised Page 4: ‘…is a peptide-centric technique [74]. In this article, we define ‘top-down proteomics’ in its broadest sense and use the term not just in a focused way describing exclusively the MS/MS analysis of an intact protein, but also include the characterization of distinct proteoforms using gel-based separation technology in combination with routine proteomic identification methods. This review includes therefore bioanalytical approaches that initiate the proteomic analysis with a separated and thus isolated proteoform of a particular protein and then employ differing MS methods for protein/peptide identification and/or PTM analysis’.

Comment 2: ‘2) In the procedure shown in Figure 2, there is a s "Top-down proteomics" step before "MS/MS analysis". However, it is not clear what this step is. Is MS/MS analysis performed after digestion? If so, isn't it just mean in-gel digestion? In addition, is it really possible to perform "Absolute mass determination"?’.

Response: In response to the reviewer’s comment, we have modified this figure, now re-numbered as Figure 4. We have changed the term ‘absolute’ to ‘protein’, so that it now reads ‘protein mass determination’ instead of ‘absolute mass determination’. We have also added the term ‘In-gel digestion’ between ‘Top-down proteomics’ and MS/MS analysis’. We hope that these changes address the reviewer’s concern.

In addition to this point and other comments by Reviewer 3, we have also modified Figure 1 and substituted the term ‘intact protein-centric proteomics’ with the term ‘proteoform-centric proteomics’ on the top of the left panel, substituted the term ‘intact muscle proteins’ with the more suitable term ‘distinct proteoforms’ at the step between separation and MS analysis, as well as changed the term ‘absolute’ to ‘protein’, so that it now reads ‘protein mass determination’ instead of ‘absolute mass determination’ in revised Figure 1. 

Comment 3: ‘Minor points, 3) Including a 2D-gel image showing muscle protein heterogeneity in this review would be helpful to understand the usefulness of 2D-PAGE in the study of skeletal muscles’. 

Response:We agree and have added new figures (Fig. 3, Fig. 5 and Fig. 6; whereby old Fig. 2 is now re-numbered as Fig. 4) to the revised manuscript with representative images of gel-based analyses of skeletal muscle samples, as well as added comments and references on 2DGE databases.

New Figures:

Figure 3. Comparison of gel electrophoretic images using 1DGE versus 2DGE analysis of contractile proteins from rodent skeletal muscle. The 1D gel shows the separation of isolated myofibrils from slow versus fast muscles and the relative position of key contractile proteins from the thick and thin filaments. The MS analysis of 1D gel zones has established the presence of isoforms of myosin heavy chain, myosin light chain, actin, troponin and tropomyosins. The 2DGE image of a separated skeletal muscle extract and its enlargement highlights the MS-based identification of proteoforms of myosin light chain (slow MLC2/P51667 and fast MLC2/P97457).

Figure 5. Overview of fluorescence two-dimensional difference in-gel electrophoresis (2D-DIGE) techniques for the comparative analysis of proteoforms. The upper panel shows the two main DIGE approaches for the systematic pre-electrophoretic labelling of protein fractions using fluorescent CyDyes. These methods utilize minimal sub-stochiometric labelling of assessable lysines in proteins or saturation labelling of assessable cysteines in proteins. The lower panel outlines 3-dye (Cy2, Cy3, Cy5) versus 2-dye (Cy3, Cy5) DIGE labelling methodology.

Figure 6. Fluorescence two-dimensional difference in-gel electrophoresis (2D-DIGE) of human gastrocnemius muscle. The 2DGE images on the left show Cy3 (Sample A) versus Cy4 (Sample B) labelled and separated muscle proteins. The densitometric analysis of the marked 2D-DIGE Zones 1 to 3 is shown on the right and depicts clusters of mostly unaltered proteins (Zone 1 and 2) versus significantly altered expression levels (Zone 3). The routine identification of distinct proteoforms is performed by MS analysis of individual 2D spots from Coomassie Brilliant Blue (CBB) stained pick gels.

Reviewer 4 Report

Kay Ohlendieck and colleagues prepared a nice, short review on the application of proteomics to studies of muscle physiology. The manuscript clearly demonstratates advantages of gel-based proteomics and top-down approach, however, it completely ignores the downsides of such approach. First of all, the top-down proteomics does not deliver reliable data on concentrations of studied proteins. Another weaknes of the approach is that many of gel-derived bands and hence, hypothetical protein species according to the top-down approach (e.g. the species which are suggested to function in a cell in their truncated forms or to differentially translated because of alternative splicing) do not represent the real functional protein forms in a living cell, but are an effect of protein disruption and/or modification during prolonged sample preparation. Thus, my suggestion is to add a short subchapter describing the weakness of such top-down approach as compared to the bottom-up proteomics.

Author Response

Response to comments by Reviewer 4:

We would like to thank Reviewer 4 for the constructive criticism of our review article. In conjunction with the comments by the other 3 reviewers, we have revised on our manuscript accordingly. This included the addition of new figures with representative gel images, including the 2D-DIGE technique, and additional discussions of technical and biological issues of gel-based proteomics, as well as more detailed descriptions of certain aspects of skeletal muscle proteomics.

Please find below point-by-point responses to individual comments by Reviewer 4:

Comment 1: ‘Kay Ohlendieck and colleagues prepared a nice, short review on the application of proteomics to studies of muscle physiology. The manuscript clearly demonstratates advantages of gel-based proteomics and top-down approach, however, it completely ignores the downsides of such approach. First of all, the top-down proteomics does not deliver reliable data on concentrations of studied proteins. Another weaknes of the approach is that many of gel-derived bands and hence, hypothetical protein species according to the top-down approach (e.g. the species which are suggested to function in a cell in their truncated forms or to differentially translated because of alternative splicing) do not represent the real functional protein forms in a living cell, but are an effect of protein disruption and/or modification during prolonged sample preparation. Thus, my suggestion is to add a short subchapter describing the weakness of such top-down approach as compared to the bottom-up proteomics.’.

Response: To address this point, we have added these critical points in the revised version of our manuscript as a new figure (Fig. 2) and revised text, as follows:

New Figure 2. Overview of the bioanalytical advantages versus potential technical limitations of gel-based top-down proteomics, especially in relation to 2DGE approaches.

Technical limitations (listed as panel in new Fig. 2):

-       Gel-based top-down proteomics is associated with prolonged periods of sample preparation and complex isolation steps, which may cause protein modifications prior to MS analysis

-       Both 1DGE and 2DGE-based top-down proteomic approaches may under-represent certain types of protein

-       Routine 2DGE does usually not properly separate high-molecular-mass proteins, highly hydrophobic proteins, low-abundance proteins and proteoforms with extremely low or high pI-values

-       Potential analytical limitations of 2DGE approaches may occur in the case of the restricted separation of complex mixtures of proteoforms, technical issues with gel-to-gel variations, the cross-contamination of individual 2D spots through highly abundant protein species, and the heterogeneous composition of large or partially overlapping 2D spots

New text added on Pages 4-5: ‘Gel-based methods are robust, highly reproducible and reliable protein separation techniques that are ideally suited for large-scale surveys of proteoforms. Staining of gel electrophoretically separated proteins can be achieved with a wide range of highly sensitive dyes, such as colloidal Coomassie Brilliant Blue, silver stains and various fluorescent dyes, as extensively reviewed [75-80]. The analytical strength of gel-based methods, as compared to protein separation by liquid chromatography, is the direct visualization of proteins within 1D gel bands or 2D gel spots. Thus, the position of discrete protein spots in 2D gels provides a swift and reliable way for determining the characteristic combination of the relative molecular mass and the pI-value of a specific proteoform of interest. Potential bioanalytical complications of gel-based top-down proteomics are associated with the fact that prolonged periods of sample preparation and complex isolation steps may introduce protein modifications prior to MS analysis. Routine 2DGE analyses may also under-estimate the presence of certain types of proteins, such as high-molecular-mass proteins, very hydrophobic proteoforms and low-abundance proteins [42]’.